# Adolescent Pregnancy in South Asia: A Systematic Review of Observational Studies

**DOI:** 10.3390/ijerph192215004

**Published:** 2022-11-15

**Authors:** Samikshya Poudel, Husna Razee, Timothy Dobbins, Blessing Akombi-Inyang

**Affiliations:** School of Population Health, University of New South Wales, Sydney, NSW 2052, Australia

**Keywords:** adolescent pregnancy, teenage pregnancy, South Asia, systematic review

## Abstract

Adolescent pregnancy is a major health and social concern in South Asia. The aim of this study is to systematically review evidence on the factors associated with adolescent pregnancy in South Asia. This study was conducted using Preferred Reporting Items for Systematic reviews and Meta-Analyses (PRISMA) 2020 guidelines. Four electronic databases: EMBASE, PubMed, CINAHL, and Scopus were searched for relevant studies on factors associated with adolescent pregnancy in South Asia published in English between January 2000 and July 2022. The quality of the included studies was assessed using 12 criteria from The National Institute of Health (NIH) Study Quality Assessment Tools for observational studies. Of the 166 articles retrieved, only 15 studies met the eligibility criteria and were included in the final analysis. Consistent factors associated with adolescent pregnancy in South Asia were low maternal education, low socioeconomic status, rural residency, and ethnic minorities. To prevent adolescent pregnancy in South Asia, concerted effort towards promoting health equity by addressing the predisposing factors associated with adolescent pregnancy is essential. This systematic review was registered with PROSPERO [CRD42022340344].

## 1. Introduction

According to the World Health Organisation (WHO), adolescent pregnancy (also known as teenage pregnancy) is pregnancy in women aged 10–19 years [1]. Adolescent pregnancy contributes significantly to the global burden of maternal and child health related morbidities and mortalities with a substantially high proportion occurring in low- and middle-income countries (LMICs). Each year, about 21 million adolescent girls become pregnant in developing countries—of which 12 million give birth [2]. Furthermore, about 10 million of these pregnancies were unplanned, thus having a significant impact on maternal and child health outcomes. Unsafe abortion, pregnancy induced hypertension, puerperal endometriosis, and eclampsia are often linked with adolescent pregnancy, and these are considered the leading causes of mortality among adolescent girls [2,3,4]. Research has shown that babies born to adolescent mothers are more likely to be of low birth weight [5], premature [6], have congenital anomaly [7], have fatal neonatal issues [8,9,10] as well as have a high propensity towards being stillborn or die within the first 7 days of life [2].

South Asia has one of the highest rates of adolescent pregnancy globally [8]. Within the region, Bangladesh, Nepal and India have reported the highest prevalence of adolescent pregnancy at 35%, 21% and 21%, respectively [9]. The high burden of adolescent pregnancy in South Asia could be due to a myriad of factors related to low socio-economic status [11,12] and a lack of comprehensive sexuality education. These are supported by the traditional social norms which encourage early marriage [10] and the low autonomy of adolescent girls.

There has been a significant global and regional effort aimed at reducing adolescent pregnancy. WHO, in collaboration with the United Nations Children’s Fund (UNICEF), United Nations Population Fund (UNFPA), and United Nations Women and Family Planning 2020, had rolled out programs to end child marriage and reduce adolescent childbearing [2]. However, these programs do not seem to adequately address the problem with 95% of total adolescent childbearing currently reported in developing countries, including South Asia [13]. To reduce adolescent pregnancy and prevent adverse maternal and child health outcomes, there is need for evidence-based adolescent reproductive health policies and strategies which address identified factors associated with adolescent pregnancy.

There have been individual studies reporting the burden and factors associated with adolescent pregnancy in various countries within South Asia [11,12,14,15]. These individual studies differ in design and methodology, thus making it difficult to make informed intra-regional comparisons and roll out effective initiatives to substantially reduce adolescent pregnancy within the region. Previous reviews conducted on adolescent pregnancy in LMICs including South Asia have identified socio-economic factors, cultural and family structure to be associated with adolescent pregnancy [9,13]. However, no study has systematically assessed the factors associated with adolescent pregnancy through a social determinants of health lens. Understanding how the social determinants of health serve as mediating factors which negatively affects adolescent pregnancy is paramount in preventing adolescent pregnancy in resource-poor settings. Hence, this review will critically analyze the factors associated with adolescent pregnancy within South Asia to inform policy direction.

## 2. Materials and Methods

### 2.1. Outcome Variable

The outcome variable for this review is adolescent pregnancy, defined as pregnancy in women aged 10–19 years [1].

### 2.2. Exploratory Variable

The modified World Health Organisation (WHO) conceptual framework [16] for social determinants of health was considered as the more structured and organized framework proposed for the selection of exploratory variables for this study. According to the framework, the variables were grouped into four main categories: structural and political context, community level factors, individual level factors and individual level behavioral and social condition (Figure 1).

### 2.3. Search Strategy

This review was conducted using the Preferred Reporting Items for Systematic reviews and Meta-Analyses (PRISMA) 2020 guidelines [17]. To identify relevant studies, a search of four electronic databases: EMBASE, PubMed, CINAHL, and Scopus was conducted using relevant MeSH words and sub-headings of keywords. The following combination of keywords was used in the search:Factor* OR determinant* OR predictor* OR risk*ANDTeenage pregnanc* OR adolescent pregnanc*ANDSouth Asia OR India OR Nepal OR Bangladesh OR Afghanistan OR Sri Lanka OR Bhutan OR Maldives OR Pakistan.

To identify any additional relevant studies that might have been missed, a search of the bibliographic references of all retrieved articles that met the inclusion criteria, complemented by citation tracking using Google Scholar, was conducted.

### 2.4. Inclusion and Exclusion Criteria

Studies were included if they (i) assessed factor/s associated with adolescent pregnancy; (ii) were conducted in South Asia including Nepal, India, Bangladesh, Sri Lanka, Bhutan, Afghanistan, Pakistan and Maldives; (iii) were published between January 2000 and July 2022; (iv) were observational studies (qualitative studies, case studies, books, policy briefs or theses were excluded); (v) were published in a peer-reviewed journal (non-peer reviewed research, review or commentaries were excluded); and (vi) were written in English. The year 2000 was used as a baseline for this review due to the significant effort South Asia has employed to prevent child marriage and promote reproductive health among adolescents. In addition, following the end of the millennium development goals (MDGs) and the introduction of the sustainable development goals (SDGs), adolescent health in South Asia has become an important public health action agenda.

### 2.5. Data Extraction

All retrieved studies were exported into an EndNote library, and duplicate records were removed. This was followed by an initial screening phase, with articles screened based on their titles and abstracts. In the final screening phase, full texts of retrieved articles were read for relevance and articles which met the inclusion criteria were retained. Data extraction and appraisal were independently conducted by two reviewers, and all disagreements between the reviewers were resolved through discussion and consensus. The summary of the selected studies was recorded including author, year, country, study design, sample size, factors associated with adolescent pregnancy, limitation and quality assessment score (Table 1).

### 2.6. Quality Appraisal

The National Institute of Health (NIH) Study Quality Assessment Tools for observational studies [29] was used to appraise the quality of the reviewed articles. The tool consists of 12 criteria to assess case-control studies and 14 criteria to assess cross-sectional and cohort studies. Scores assigned to case-control studies were 0–12, whereas, for cross-sectional and cohort studies, were 0–14 [0 if none of the criteria were met and 12 or 14 points if all criteria were met]. The sum of points awarded represented the overall quality score of a study. Studies were rated as poor quality (score ≤ 4); medium quality (5–9); and high quality (≥10) as shown in Appendix A.

## 3. Results

A total of 166 articles were retrieved from four bibliographic databases and Google Scholar. After the entire screening process, 15 studies were retained for the final analysis as shown in Figure 2.

### 3.1. Characteristics of the Included Studies

Table 1 shows a summary of the studies included in this review. Of the 15 retained studies, 1 was cohort, 3 were case-control and 11 were cross-sectional studies. Six were conducted in Bangladesh, 6 in Nepal, 2 in Sri Lanka and 1 in Pakistan. No studies were conducted in India, Afghanistan, Bhutan and the Maldives. The number of samples included in 15 studies ranged from 140 to 400,076 participants. Based on the 12 criteria used to evaluate the quality of case-control studies, all three included case-control studies were of medium quality. Likewise, based on the 14 criteria to evaluate the quality of cross-sectional and cohort studies, none of the studies were of high quality, 11 were of medium quality, while one was of low quality. The details of domain-specific score are provided in Appendix A.

### 3.2. Evidence from Reviewed Studies

The level of education attainment was found to be the most consistent factor associated with adolescent pregnancy. Studies conducted in Nepal [11,23,24,25,27], Pakistan [14] and Bangladesh [12,18,19,20,21] reported an association between lower level of education and adolescent pregnancy. Studies conducted in Bangladesh [19] and Pakistan [14] also reported that a partner’s lower education level was associated with a higher rate of adolescent pregnancy. However, studies conducted in Bangladesh [12,18] found that a partner’s primary or secondary education had a significant association with an increased rate of adolescent pregnancy compared to their uneducated peer.

Eight studies conducted in Bangladesh [12,18,19,20,21], Nepal [11,27] and Pakistan [14] reported higher odds of adolescent pregnancy among women of lower economic status. Three cross-sectional studies from Bangladesh [18,19,20] and a case-control study from Sri Lanka [28] showed a higher propensity of adolescent pregnancy among Muslims than other religions. Five cross-sectional studies from Bangladesh [19,20,21,22] and Pakistan [14] examined the impact of place of residence on adolescent pregnancy and found a higher rate of adolescent pregnancy among women residing in rural areas compared to those in urban areas. In addition, four studies from Bangladesh [18,19], Pakistan [14] and Nepal [24] reported that unemployed women had a higher propensity towards adolescent pregnancy. However, in contrast, a study conducted in Bangladesh [20] revealed that employed woman were more predisposed to adolescent pregnancy. Three studies conducted in Bangladesh [18,19,21] reported that women residing in Raj Shahi division were at a greater risk of adolescent pregnancy. Similarly, a study conducted in Pakistan showed that women residing in Sindh, KPK and Baluchistan provinces were more prone to adolescent pregnancy [14].

Studies conducted in Bangladesh [12,18] and Nepal [25] found early marriage to be associated with adolescent pregnancy. Studies conducted in Nepal [11,24,26] reported a higher number of adolescent pregnancies among girls from disadvantaged ethnic groups such as Dalit and Madhesi. Apart from a Nepalese context, a case-control study from Sri Lanka [28] also found that women from a Tamil ethnic background were more prone to adolescent pregnancy compared to those of non-Tamil ethnicity. A cohort study from Sri Lanka [15] revealed that lower contraceptives use was significantly associated with adolescent pregnancy; in contrast, a cross-sectional study from Bangladesh [18] found that women who used contraceptives were more predisposed to adolescent pregnancy as compared to those who never used contraceptives. Studies in Nepal [11] and Bangladesh [19,21] found a higher level of adolescent pregnancies among girls with no access to mass media. Advanced spousal age was reported to be associated with adolescent pregnancy among married Bangladeshi [12,19] and Nepalese [26] women. Furthermore, individual behavior such as women with the habit of alcohol and tobacco use were more likely to report adolescent pregnancy in Nepal [23,25,26].

## 4. Discussion

This systematic review highlights important factors associated with adolescent pregnancy in South Asia. The most consistent factors associated with adolescent pregnancy in South Asia include low education attainment, poor economic status, rural residence and women from a Muslim religious background.

A lower education level of adolescent women as well as their male counterparts was reported as one of the most consistent factors associated with adolescent pregnancy across the countries in South Asia. Previous research suggested that educated men and women are better informed about protective sexual intercourse particularly contraception as well as the adverse effects of teenage childbearing and its consequences in family and community [30,31,32]. Despite coordinated efforts from government and multi-lateral organizations to provide equal and quality education, South Asia still has the highest youth population with low education attainment [33]. However, contrary to this finding, few studies from Bangladesh using Bangladesh Demographic and Health Survey data found adolescent pregnancy to be higher among girls and their male partners with primary and secondary education compared to uneducated girls and their partners.

Low economic status was reported to be associated with adolescent pregnancy in South Asia. Socio-economic factors including access to education, occupation and income generating programs are very essential to ensure health and wellbeing of adolescent girls. This could assist girls with avoiding early pregnancy or make them better equipped to manage if they do become pregnant [34]. With soaring unemployment rates resulting in low income, South Asia is home to considerably low socio-economic population [35]. Unaffordable self-financed health care services in some countries like India, Nepal, and Bangladesh create challenges in accessing quality reproductive health care services including contraceptives for adolescents with lower socioeconomic status [36,37,38].

In this review, women residing in rural areas were found to be more susceptible to adolescent pregnancy. In addition, predominantly rural divisions/provinces with adolescents of low economic status in Bangladesh and Pakistan reported a greater rate of adolescent pregnancy. Consistent with our findings are similar studies conducted in Africa [39] and other LMICs [13]. Adolescent girls residing in urban areas are generally more educated and socioeconomically better off than those residing in rural areas [40]. Due to economic challenges and barriers to education and early marriage, adolescent girls residing in rural areas tend to have a greater predisposition towards pregnancy [19,41,42].

Religion in South Asia is a medium for socialization which shapes the ideology within the community by impacting health, mainly women’s health, in many ways. In South Asia, the most common religions are Hinduism, Islam, Buddhism and Christianity with each religion having unique values and beliefs. Research has shown an association between Muslim religion and early marriage [40,43,44], which could result in adolescent pregnancy. One of the plausible reasons for this is the belief that Islam promotes early marriage stipulating no age limits for girl marriage.

From a study conducted in South Africa, it was inferred that not only are young woman reluctant to use contraceptives to prove their relationship commitment to their male counterparts, but they are also pressured or have their pills thrown out by their partners, which was indicative of power dynamics within relationships, and thus get exposed to untimely pregnancies [45]. In addition, young girls also face a barrier in accessing contraceptives due to restrictive government policies, lack of autonomy, inaccessibility due to poor transportation and lack of knowledge [14,46].

This review is a comprehensive search of literature which followed PRISMA guidelines, and the quality appraisal of reviewed articles was conducted using NIH study quality assessment tools for observational studies to ensure the methodological rigor. Furthermore, predefined MeSH words and sub-headings of keywords applied into four electronic databases along with a thorough screening of identified studies by two authors increased robustness in search strategy and the study selection process. However, this review did not conduct a meta-analysis to estimate the burden and factors associated with adolescent pregnancy in South Asia, and the conclusion was drawn based on narrative synthesis. This was mainly due to the discrepancy in definition used for adolescent pregnancy for which the estimation from the meta-analyses would be subject to bias. This review did not find studies from India, Afghanistan, Bhutan and the Maldives that met the inclusion criteria. Future research on adolescent pregnancy in these countries is recommended to fully understand its associated factors within the region.

## 5. Conclusions

This systematic review identified important factors associated with adolescent pregnancy in South Asia. The findings suggest that the most common factors associated with adolescent pregnancy in South Asia include low education attainment, poor economic status and rural residence, belonging to ethnic minorities and those having a Muslim religious background.

To prevent adolescent pregnancy in South Asia, there is a need for regional effort towards promoting girl child education, mainly focusing on rural locations and ethnic minorities. The implementation of effective policies around girls’ education and income generating vocational initiatives would aid in reducing pregnancies among adolescents with low socioeconomic background and subsequently assist the region with achieving the Sustainable Development Goal targets around the prevention of maternal death, and the promotion of free, equitable and quality primary and secondary education for girls.

## Figures and Tables

**Figure 1 ijerph-19-15004-f001:**
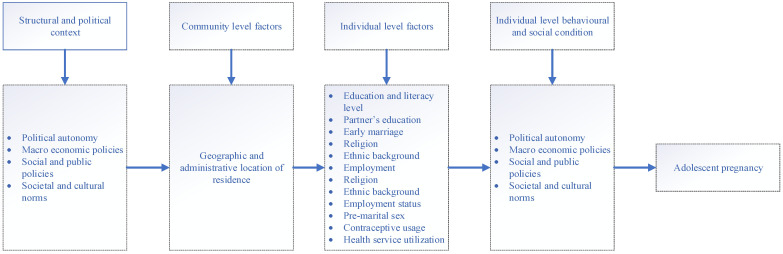
Conceptual framework for social determinants of adolescent pregnancy.

**Figure 2 ijerph-19-15004-f002:**
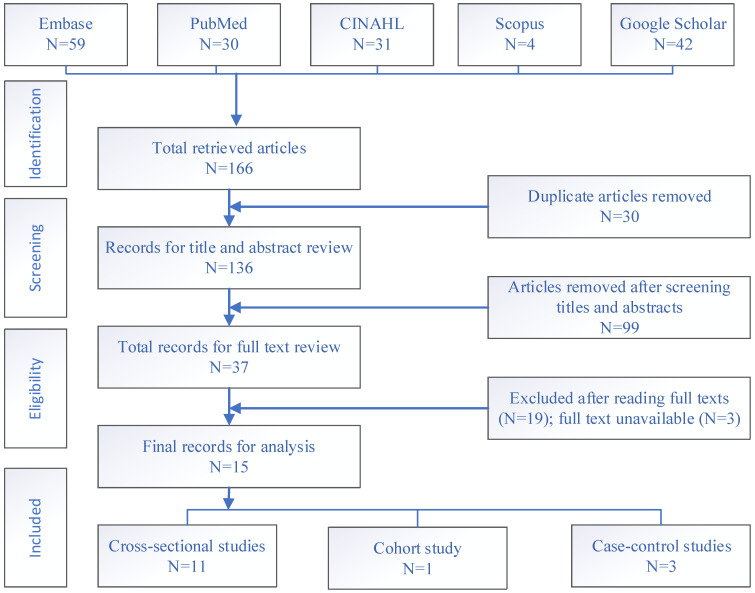
Flow chart for study selection based on PRISMA 2020 guidelines.

**Table 1 ijerph-19-15004-t001:** Summary of the selected studies.

Author, Year, Country [REF]	Study Design	Sample Size (*N*)	Factors Associated with Adolescent Pregnancy	Limitation	Quality Assessment Score
Ali et al., 2020 Bangladesh [18]	Cross-sectional	*N* = 15,842 Ever married women	Risks: Primary and secondary education, unemployment, Primary education of husband, Muslim Religion, less than 18 years of marriage (early marriage), use of contraceptives. Protective: Higher education, employment, higher wealth index.	Possibility of recall bias because the responses was retrospectively collected, and it was difficult to establish a causal relationship between adolescent pregnancy and its associated risk factors due to its cross-sectional nature.	8 Medium
Sarder et al., 2020 Bangladesh [19]	Cross-sectional	*N* = 4608 Adolescent married women	Risks: Rangpur, Raj Shahi, Chittagong, Khulna, Dhaka Barisal resident; rural resident, women with primary education or no education, spouse with primary education or no education, women from joint families, poor and middle-class women, women with Muslim faith, higher spousal age gap, unemployed women, no access to mass media	The sample was limited to ever married teen aged women. As the study used BDHS 2014, data collection is subjected to recall bias due to its retrospective nature. Difficulty with establishing causal inferences between the factors and outcome variable due to its cross-sectional nature.	8 Medium
Alam et al., 2018 Bangladesh [12]	Cross-sectional	*N* = 7641 Adolescent girls	Risks: Primary and secondary educated woman compared to women with no education, husband education: Primary and secondary education, age at marriage ≤ 15 years, low socio-economic status, Rangpur region resident. Protective: Higher education level, age of marriage ≥ 15 years	Difficulty with establishing a causal relationship between adolescent pregnancy and its associated risk factors due to its cross-sectional nature. Nationally representative sample size with recall period of 5 years may be subject to respondent recall bias.	8 Medium
Haq et al., 2018 Bangladesh [20]	Cross-sectional	*N* = 17,863 Ever married women	Risks: Woman residing in Khulna, Raj Shahi, Rangpur and Barisal compared to Chittagong, Sylhet, Dhaka division; rural resident, Muslim religion, Poor economic condition, employed woman, woman with no formal education, husband with no formal education.	The sample was limited to ever married women, so the missing out of unmarried women and their pregnancy history is possible. Subject to recall bias due to retrospective nature of data collection. Difficulty with establishing a causal relationship between independent variables and outcomes due to study design.	7 Medium
Islam et al., 2017 Bangladesh [21]	Cross-sectional	*N* = 6608 Adolescent girls	Risks: Poorest wealth quantile, rural resident, woman’s no education, Raj Shahi division. Protective: Lower spousal age gap, exposure to media.	Multivariate Analysis was performed only on BDHS 2004–2014 data rather than all the previous data. Adolescent motherhood classification (proximal and distal) not shown in the study. Recall bias and difficulty establishing a causal relationship between exposure and outcome variable.	8 Medium
Sayem et al., 2011 Bangladesh [22]	Cross-sectional	*N* = 389 Married women	Desired >2 children, participants aged 20–24 years, rural resident.	Due to a convenience sampling technique, finding can be highly unrepresentative and cannot be generalisable to woman except young and sampled woman. In addition, because of the cross-sectional study design, it was unable to establish a causal relationship between the factors and adolescent pregnancy.	7 Medium
Gurung et al., 2020 Nepal [23]	Cross-sectional	*N* = 60,742 women	Risks: Disadvantaged ethnic group, no formal education.	Possible recall and interviewer bias due to retrospective nature of data collection. Interviewer was unable to interview mothers of stillbirth babies which might have resulted in the missing out of important information and resulted in biased outcome.	3 Poor
Neupane et al., 2019 Nepal [24]	Case-control	*N* = 432 respondents	Risks: Disadvantaged ethnic group, agricultural occupation, labour occupation, education level ≤ secondary level of education, unemployed women, unplanned pregnancy.	Despite pre-planned structured interviews, these interviews from respondents are prone to recall bias on the responses regarding exposure variables.	6 Medium
Poudel et al., 2018 Nepal [11]	Cross-sectional	*N* = 7788 Adolescent women	Risks: Middle household wealth index, poor household wealth index, Dalit, Madhesi, Unemployed. Protective: Educated women, woman with access to media exposure to public health issues.	Information provided by the respondent was based on self-report, which may result in recall bias. Analysis was based on cross sectional data, and hence the establishment of clear temporal relation between study outcome and confounding factors cannot be determined.	8 Medium
Devkota et al., 2018 Nepal [25]	Cross-sectional	*N* = 454 women	Risks: Alcohol consumption. Protective: Secondary level education, married after 17 years, Fairs/clubs attended.	Sampling of participants only included those women attending health facilities and the missing out of non-attendants might have produced biased results. Cross sectional nature limits the ability to establish causal direction between independent and dependent variables. Study lacked information on socio-cultural aspects of women, which may have huge influence on adolescent childbearing behaviour. The possibility of under reporting of sexual behaviour by participants due to its culturally sensitive nature might have biased the result.	6 Medium
Pradhan et al., 2018 Nepal [26]	Cross-sectional	*N* = 2524 Married women	Risks: Living in the eastern development region, sexual intercourse at early age, women who had an older husband, 2006 and 2011.	The inclusion of two important explanatory variables (use of contraception and exposure to mass media) in the analysis section was not possible due to NDIS questionnaire format. The sample only included married women, so the missing out of unmarried adolescent pregnancy is possible.	8 Medium
Sharma et al., 2002 Nepal [27]	Case-control	*N* = 140 respondents	Risks: low education status, Poor economic background, love marriages.	The study failed to conduct multivariable analysis, reaching the conclusion based on univariate analysis, which might not be sufficient to provide an actual relationship between the outcome variable and exploratory variables.	7 Medium
Agampodi et al., 2021 Sri Lanka [15]	Cohort	*N* = 3370 women	Risks: Maternal-Paternal low education level, low contraceptive use before pregnancy, being unmarried, less time since marriage, low utilization of pre-conceptional health care services.	Due to the cohort nature, the undetected pre outcome might have affected the exposure factor prior to the actual outcome (adolescent pregnancy). Despite high coverage of the maternal care program, the missing out of a few mothers registered in the program is possible. In addition, the author fears that the service provision might not have been delivered in a similar way within the district, due to which the pre conceptional care service utilization should be carefully analysed.	9 Medium
Dulitha et al., 2013 Sri Lanka [28]	Case-control	*N* = 510 teenagers	Risks: Tamil and Muslim religion, lack of knowledge of disadvantage of teenage pregnancy, less support from teachers, lack of strictness in family, higher confidence in decision-making.	Due to its case-control nature, recall bias from the respondent while gathering responses on factors and adolescent pregnancy is possible.	6 Medium
Ali et al., 2021 Pakistan [14]	Cross-sectional	*N* = 400,076 Ever married pregnant women	Risks: Rural resident, Sindh, KPK, Baluchistan, poor wealth Index, no/lower woman and partner education, not working or unskilled occupation, no access to knowledge of family planning.	Difficulty with establishing a causal relationship between adolescent pregnancy and its associated risk factors due to its cross-sectional nature. Possible recall bias due to self-reported questionnaire used for collecting information on a majority of factors. Data collection did not cover all the clusters in each DHS due to security issues and non-participation. In addition, some of the states and provinces like Azad Jammu and Kashmir, Federally Administered Tribal Regions and Gilgit and Baltistan regions were not covered in all PDHS except for the latest one and were merged with the other region in the former one.	8 Medium

## Data Availability

This is a systematic review of published research which is available online.

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
