# Peer review of "Adolescent Pregnancy in South Asia: A Systematic Review of Observational Studies"

_ijerph, 2022, doi:10.3390/ijerph192215004_

Round 1

Reviewer 1 Report

The discussion of findings of the review is limited. This can be expanded to include how the issues intersect. 

Similarly, the conclusion is very brief. It could be expanded to include how the sustainable goals could be addressed. 

Author Response

12/11/2022

To

Angkana Jantanaprasartporn

Section: Global Health

Subject: Revision of manuscript titled “Adolescent Pregnancy in South Asia: A Systematic Review of Observational Studies”

IJERPH Editorial Office MDPI,

St. Alban-Anlage 66, 4052 Basel, Switzerland

Dear Editor,

On behalf of all co-authors, I would like to thank the reviewer for the important comments, and suggestions on our manuscript titled “Adolescent Pregnancy in South Asia: A Systematic Review of Observational Studies” (Manuscript ID: ijerph-1937276). We are submitting the revised version of the manuscript for your consideration. Responses to reviewer-1 comments, and details of the changes that are made into the revised version are provided below.

Comment: The discussion of findings of the review is limited. This can be expanded to include how the issues intersect.

Response: Thank you for your comment. It is hard to ascertain how board you would like our findings to be discussed. We believe our discussion is exhaustive and have adequately highlighted areas of possible intersections identified in our study.

Comment: Similarly, the conclusion is very brief. It could be expanded to include how the sustainable goals could be addressed.

Response: Thank you for your comment. We have revised the conclusion section in-line with your comment. The following lines have been added:

“…and subsequently assist the region achieve Sustainable Development Goal targets around prevention of   maternal death; and promotion of free, equitable and quality primary and secondary education for girls”.

We believe we have appropriately addressed all the comments from the reviewer, and that the

revised manuscript will be considered for publication.

Kind regards

Samikshya Poudel (Corresponding Author)

samikshya.poudel@unsw.edu.au

School of Population Health, University of New South Wales

Sydney, NSW 2052, Australia

Reviewer 2 Report

Thank you for the opportunity to review this paper. It is an important piece of work that will make a valuable contribution to efforts to reduce adolescent pregnancy in South Asia and the broader Asia-Pacific region. The paper was well-written and clear.

I have some minor suggestions/comments for the authors to consider:

Introduction

Line 42: suggest changing "estimate" to "prevalence" to be more specific?

Line 50: please add "s" to "United Nation" or spell out "United Nations Entity for Gender Equality and the Empowerment of Women (UN Women)" for consistency with UNICEF and UNFPA also mentioned in the sentence

Materials and Methods

Line 107-109: Using the year 2000 as a baseline makes a lot of sense as it aligns with the date of signing of the UN Millennium Declaration, but as it is currently written, this sentence seems a little out of place here/might be better in the introduction.

Results

Line 140-141: I find it quite interesting that the systematic review included/found no studies conducted in India. Given the size of the population in India, and the proportion who are young people who may be at risk of adolescent pregnancy, I think this is an important limitation of the scope. Could the authors share some insight as to why this might be?

Line 145: I also find it interesting that none of the included studies were of high quality.

Line 153: Would be good to clarify here the direction of the association - was partner education positively or negatively associated with adolescent pregnancy?

Line 158: Did these studies check for possible correlation between low SES and level of education/educational attainment?

Line 167-170: Might be good to include here (or maybe in discussion) some information on the significance of these findings -- e.g., are these divisions/provinces of lower economic status?

Line 188: missing "with" (associated with)

Line 197-199: Suggest removing comma after "Despite". Also, it might be good to cite here other (government/non-government) actors that UNICEF works with to help improve the provision of equal and quality education in South Asia, or to make it more generic (e.g., despite coordinated efforts from government and multilateral organisations...) to acknowledge that the persistence of low educational attainment is not just UNICEF's responsibility.

Line 204-206: As argued by Ahorlu et al. (2015, p.1), "adolescent girls, especially those that get pregnant should not be viewed as weak and vulnerable". In that context, it might be good to consider replacing the term "deprivation" with a more neutral word, or rephrasing this sentence to return to Ahorlu et al.'s key finding that having access to social, economic, and cultural capitals can help girls to avoid adolescent pregnancy or make them better equipped to manage if they do become pregnant (rather than the absence of these necessarily leading/contributing to early marriage or pregnancy).

Line 210: "deprived" - as above

Line 214-216: The study area Dizon-Luna's work (San Pablo City in the Philippines) is an urban area and is probably not the best supporting reference to this point.

I believe this is statement might be better supported by the following:

Gayatri, M. (2021). Socioeconomic determinants of adolescent pregnancy in Indonesia: A population-based survey. Annals of Tropical Medicine and Public Health, 24, 24-186.

Sarder, M. A., Alauddin, S., & Ahammed, B. (2020). Determinants of teenage marital pregnancy among Bangladeshi women: An analysis by the Cox proportional hazard model. Social Health and Behavior, 3(4), 137.

Sychareun, V., Vongxay, V., Houaboun, S., Thammavongsa, V., Phummavongsa, P., Chaleunvong, K., & Durham, J. (2018). Determinants of adolescent pregnancy and access to reproductive and sexual health services for married and unmarried adolescents in rural Lao PDR: a qualitative study. BMC Pregnancy and Childbirth, 18(1), 1-12.

Line 215: Suggest rephrasing "high school dropouts" to draw attention to the cause rather than the outcome (e.g., "barriers to education" and not the girls who drop out of high school are contributing to increased risk in rural areas).

Line 220: add "s" to "belief"

Line 221-222: Given the first point in this sentence, it might be good to cite here literature that also links child marriage with adolescent pregnancy. You may find this reference useful (it cites many research studies from Asia Pacific that might be relevant):

UNFPA. (2021). My Body is My Body, My Life is My Life: Sexual and reproductive health and rights of young people in Asia and the Pacific. Retrieved from Bangkok, Thailand: https://asiapacific.unfpa.org/en/publications/my-body-my-body-my-life-my-life-sexual-and-reproductive-health-and-rights-young-people

Line 222-224: I understand that the association of adolescent pregnancy and identification with Islam was a notable finding of the systematic review. We certainly cannot deny the impact of religious values on adolescents' decision-making. However, I do wonder if this finding was strong enough to warrant this attention/elaboration, and what this point is intended to achieve. Religion and the values and beliefs it is founded on is is unlikely to be moved easily by any policy or program seeking to address adolescent pregnancy. It might be better to keep this section short (e.g., I'm not sure if the last sentence in line 223-224 necessarily strengthens your paper), especially since there is no accompanying recommendation on religion in the abstract or conclusion and policy implications section of this paper.

Line 225-227: I think a topic sentence here establishing that this paragraph is in relation to your findings on contraceptive use would strengthen the point made by the the succeeding sentences.

I would suggest rephrasing the first sentence, as the term "inappropriate physical relationships" comes across as a value judgement. It is perfectly normal for young people to have physical and sexual relationships, and I don't believe it is our place to determine what is an "appropriate" or "inappropriate" relationship.

Also, I think that it might be good to clarify that the findings of Wood and Jewkes (2006), as cited here, was not only that some young people considered pregnancy as a way to prove love and commitment (and for males, fertility), but also that even when they wanted to use (or were using) contraceptives, some girls were pressured or forced by their male partners to stop (e.g., some were verbally pressured, others had their pills thrown away by the partner). This was indicative of power dynamics within relationships - i.e., relationship level barriers to contraceptive use - which not only made girls reluctant to use contraceptives, but in essence, gave them little or no choice in the matter. I think this is a valuable point to make here, given that the next sentence speaks to barriers to accessing contraceptives.

Author Response

12/11/2022

To

Angkana Jantanaprasartporn

Section: Global Health

Subject: Revision of manuscript titled “Adolescent Pregnancy in South Asia: A Systematic Review of Observational Studies”

IJERPH Editorial Office MDPI,

St. Alban-Anlage 66, 4052 Basel, Switzerland

Dear Editor,

On behalf of all co-authors, I would like to thank the reviewer for the important comments, and suggestions on our manuscript titled “Adolescent Pregnancy in South Asia: A Systematic Review of Observational Studies” (Manuscript ID: ijerph-1937276). We are submitting the revised version of the manuscript for your consideration. Responses to reviewer-2 comments, and details of the changes that are made into the revised version are provided below.

Comment: Thank you for the opportunity to review this paper. It is an important piece of work that will make a valuable contribution to efforts to reduce adolescent pregnancy in South Asia and the broader Asia-Pacific region. The paper was well-written and clear.

Response: Thank you for your encouraging and inspiring comment.

Comment: Line 42: suggest changing "estimate" to "prevalence" to be more specific?

Response: Agreed and changed.

Comment: Line 50: please add "s" to "United Nation" or spell out "United Nations Entity for Gender Equality and the Empowerment of Women (UN Women)" for consistency with UNICEF and UNFPA also mentioned in the sentence.

Response: Agreed and revised in the manuscript.

Comment: Line 107-109: Using the year 2000 as a baseline makes a lot of sense as it aligns with the date of signing of the UN Millennium Declaration, but as it is currently written, this sentence seems a little out of place here/might be better in the introduction.

Response: This sentence was the justification for one of the inclusion criteria (published between January 2000 and July 2022). Therefore, we think it rightly positioned. 

Comment: Line 140-141: I find it quite interesting that the systematic review included/found no studies conducted in India. Given the size of the population in India, and the proportion who are young people who may be at risk of adolescent pregnancy, I think this is an important limitation of the scope. Could the authors share some insight as to why this might be?

Response: Thank you for this valid observation. We have included below text as part of study limitation.

“This review did not find studies from India, Afghanistan, Bhutan, and Maldives that met the inclusion criteria. Future research on adolescent pregnancy in these countries is recommended to fully understand its associated factors within the region.”

Comment: Line 145: I also find it interesting that none of the included studies were of high quality.

Response: Quality appraisal of reviewed articles was conducted using NIH study quality assessment tools for observational studies to ensure methodological rigor. We have no control over quality of studies included in this review as we adhered strictly to the pre-set guidelines.

Comment: Line 153: Would be good to clarify here the direction of the association - was partner education positively or negatively associated with adolescent pregnancy?

Response: Thank you for your comment. We have revised the manuscript to include the direction of the association. It now reads:

“Studies conducted in Bangladesh [19] and Pakistan [14] also reported that partner’s lower education level was associated with higher rate of adolescent pregnancy”.

Comment: Line 158: Did these studies check for possible correlation between low SES and level of education/educational attainment?

Response: Thank you for your comment. Some included studies checked for possible correlation between low SES and level of education/educational attainment. However, our study only reported consistent factors associated with adolescent pregnancy across all studies. Assessment of correlation between factors is beyond the scope of this review.

Comment: Line 167-170: Might be good to include here (or maybe in discussion) some information on the significance of these findings -- e.g., are these divisions/provinces of lower economic status?

Response: Thank you for your comment. We have revised our manuscript in-line with you comment. It now reads in the discussion section: “In addition, predominantly rural divisions/provinces with adolescents of low economic status in Bangladesh and Pakistan reported greater incidence of adolescent pregnancy”.

Comment: Line 188: missing "with" (associated with)

Response: This has been addressed. Thank you.

Comment: Line 197-199: Suggest removing comma after "Despite". Also, it might be good to cite here other (government/non-government) actors that UNICEF works with to help improve the provision of equal and quality education in South Asia, or to make it more generic (e.g., despite coordinated efforts from government and multilateral organisations...) to acknowledge that the persistence of low educational attainment is not just UNICEF's responsibility.

Response: Agreed. Manuscript has been revised to read thus:

“Despite, coordinated efforts from government and multi-lateral organizations to provide equal and quality education, South Asia still has the highest youth population with low education attainment”

Comment: Line 204-206: As argued by Ahorlu et al. (2015, p.1), "adolescent girls, especially those that get pregnant should not be viewed as weak and vulnerable". In that context, it might be good to consider replacing the term "deprivation" with a more neutral word, or rephrasing this sentence to return to Ahorlu et al.'s key finding that having access to social, economic, and cultural capitals can help girls to avoid adolescent pregnancy or make them better equipped to manage if they do become pregnant (rather than the absence of these necessarily leading/contributing to early marriage or pregnancy).

Response: Agreed and revised accordingly. The sentence now reads:” Socio-economic factors including access to education, occupation and income generating programs are very essential to ensure health and wellbeing of adolescent girls. This could assist girls avoid early adolescent pregnancy or make them better equipped to manage if they do become pregnant [34]”.

Comment: Line 210: "deprived" - as above

Response: Agreed and changed accordingly.

Comment: Line 214-216: The study area Dizon-Luna's work (San Pablo City in the Philippines) is an urban area and is probably not the best supporting reference to this point.

I believe this is statement might be better supported by the following:

Gayatri, M. (2021). Socioeconomic determinants of adolescent pregnancy in Indonesia: A population-based survey. Annals of Tropical Medicine and Public Health, 24, 24-186.

Sarder, M. A., Alauddin, S., & Ahammed, B. (2020). Determinants of teenage marital pregnancy among Bangladeshi women: An analysis by the Cox proportional hazard model. Social Health and Behavior, 3(4), 137.

Sychareun, V., Vongxay, V., Houaboun, S., Thammavongsa, V., Phummavongsa, P., Chaleunvong, K., & Durham, J. (2018). Determinants of adolescent pregnancy and access to reproductive and sexual health services for married and unmarried adolescents in rural Lao PDR: a qualitative study. BMC Pregnancy and Childbirth, 18(1), 1-12.

Response: Thank you for the important comment, and we have supported the statement with the reference provided.

Comment: Line 215: Suggest rephrasing "high school dropouts" to draw attention to the cause rather than the outcome (e.g., "barriers to education" and not the girls who drop out of high school are contributing to increased risk in rural areas).

Response: Agreed and addressed in the revised manuscript. The sentence now reads: “Due to economic challenges, barriers to education and early marriage, adolescent girls residing in rural areas tend to have a greater predisposition towards pregnancy [19, 41, 42].”

Comment: Line 220: add "s" to "belief"

Response: Added as advised.

Comment: Line 221-222: Given the first point in this sentence, it might be good to cite here literature that also links child marriage with adolescent pregnancy. You may find this reference useful (it cites many research studies from Asia Pacific that might be relevant):

UNFPA. (2021). My Body is My Body, My Life is My Life: Sexual and reproductive health and rights of young people in Asia and the Pacific. Retrieved from Bangkok, Thailand: https://asiapacific.unfpa.org/en/publications/my-body-my-body-my-life-my-life-sexual-and-reproductive-health-and-rights-young-people

Response: Agreed, and the statement has been supported by relevant article.

Comment: Line 222-224: I understand that the association of adolescent pregnancy and identification with Islam was a notable finding of the systematic review. We certainly cannot deny the impact of religious values on adolescents' decision-making. However, I do wonder if this finding was strong enough to warrant this attention/elaboration, and what this point is intended to achieve. Religion and the values and beliefs it is founded on is is unlikely to be moved easily by any policy or program seeking to address adolescent pregnancy. It might be better to keep this section short (e.g., I'm not sure if the last sentence in line 223-224 necessarily strengthens your paper), especially since there is no accompanying recommendation on religion in the abstract or conclusion and policy implications section of this paper.

Response: We agreed with your line of thought. However, we cannot deny the existing findings and thus worth highlighting. Following reviewer’s comment, we have removed lines 223-224 from the manuscript.

Comment: Line 225-227: I think a topic sentence here establishing that this paragraph is in relation to your findings on contraceptive use would strengthen the point made by the succeeding sentences.

I would suggest rephrasing the first sentence, as the term "inappropriate physical relationships" comes across as a value judgement. It is perfectly normal for young people to have physical and sexual relationships, and I don't believe it is our place to determine what is an "appropriate" or "inappropriate" relationship.

Response: Agreed, and the sentence has been removed.

Comment: Also, I think that it might be good to clarify that the findings of Wood and Jewkes (2006), as cited here, was not only that some young people considered pregnancy as a way to prove love and commitment (and for males, fertility), but also that even when they wanted to use (or were using) contraceptives, some girls were pressured or forced by their male partners to stop (e.g., some were verbally pressured, others had their pills thrown away by the partner). This was indicative of power dynamics within relationships - i.e., relationship level barriers to contraceptive use - which not only made girls reluctant to use contraceptives, but in essence, gave them little or no choice in the matter. I think this is a valuable point to make here, given that the next sentence speaks to barriers to accessing contraceptives.

Response: We agree with the reviewer. We have revised the manuscript with below text:

Study conducted in South Africa, it was inferred that not only young woman are reluctant to use contraceptives to prove their relationship commitment to their male counterparts, but also are pressured or have their pills thrown by their partners; which was indicative of power dynamics within relationship, and thus get exposed to untimely pregnancies [45]. In addition, young girls also face barrier in accessing contraceptives due to restrictive government policies, lack of autonomy, inaccessibility due to poor transportation and lack of knowledge [14, 46].

We believe we have appropriately addressed all the comments from the reviewer, and that the revised manuscript will be considered for publication.

Kind regards

Samikshya Poudel (Corresponding Author)

samikshya.poudel@unsw.edu.au

School of Population Health, University of New South Wales

Sydney, NSW 2052, Australia
